# Advanced Brain Imaging in Preterm Infants: A Narrative Review of Microstructural and Connectomic Disruption

**DOI:** 10.3390/children9030356

**Published:** 2022-03-04

**Authors:** Philippe Vo Van, Marianne Alison, Baptiste Morel, Jonathan Beck, Nathalie Bednarek, Lucie Hertz-Pannier, Gauthier Loron

**Affiliations:** 1Department of Neonatology, Hospices Civils de Lyon, Femme Mère Enfant Hospital, 59 Boulevard Pinel, 69500 Bron, France; 2Service d’Imagerie Pédiatrique, Hôpital Robert Debré, APHP, 75019 Paris, France; marianne.alison@aphp.fr; 3U1141 Neurodiderot, Équipe 5 inDev, Inserm, CEA, Université de Paris, 75019 Paris, France; lucie.hertz-pannier@cea.fr; 4Pediatric Radiology Department, Clocheville Hospital, CHRU of Tours, 37000 Tours, France; baptiste.morel@univ-tours.fr; 5UMR 1253, iB-Rain, Université de Tours, Inserm, 37000 Tours, France; 6Department of Neonatology, Reims University Hospital Alix de Champagne, 51100 Reims, France; jbeck@chu-reims.fr (J.B.); nbednarek@chu-reims.fr (N.B.); gloron@chu-reims.fr (G.L.); 7CReSTIC EA 3804, Université de Reims Champagne Ardenne, 51100 Reims, France; 8NeuroSpin, CEA-Saclay, Université Paris-Saclay, 91191 Gif-sur-Yvette, France

**Keywords:** preterm infants, microstructure, connectivity, MRI

## Abstract

Preterm birth disrupts the in utero environment, preventing the brain from fully developing, thereby causing later cognitive and behavioral disorders. Such cerebral alteration occurs beneath an anatomical scale, and is therefore undetectable by conventional imagery. Prematurity impairs the microstructure and thus the histological process responsible for the maturation, including the myelination. Cerebral MRI diffusion tensor imaging sequences, based on water’s motion into the brain, allows a representation of this maturation process. Similarly, the brain’s connections become disorganized. The connectome gathers structural and anatomical white matter fibers, as well as functional networks referring to remote brain regions connected one over another. Structural and functional connectivity is illustrated by tractography and functional MRI, respectively. Their organizations consist of core nodes connected by edges. This basic distribution is already established in the fetal brain. It evolves greatly over time but is compromised by prematurity. Finally, cerebral plasticity is nurtured by a lifetime experience at microstructural and macrostructural scales. A preterm birth causes a negative and early disruption, though it can be partly mitigated by positive stimuli based on developmental neonatal care.

## 1. Introduction

Over the past decades, a shift has been taking place in the spectrum of impairments sustained by preterm born infants (PT). Medical progress has led to a decrease of mortality, as well as severe cerebral palsy [1,2,3,4]. Meanwhile, childhood follow-up studies reveal altered cognitive outcomes, with high prevalence of executive function and neuropsychiatric disorders [4,5]. Schooling is frequently impaired, and requires special educational support. Moreover, social integration and quality of life may be affected [6].

The understanding of those disabilities has greatly benefited from brain magnetic resonance imaging (MRI). Cerebral ultrasound remains an optimal investigation in daily practice to detect intraventricular hemorrhages and cystic leukomalacia [7]. However, MRI allows a more accurate investigation of the brain. It corroborates the emergence of mild to moderate motor and cognitive disorders to the histological cerebral lesions and the concept of encephalopathy of prematurity described by J Volpe [8]. Indeed, neonatal MRI studies in the early 2000s have identified white matter (WM) signal abnormalities as nodular periventricular leukomalacia in favor of gliosis lesions and predictive of neurodevelopmental outcomes [1,9,10]. Therefore, cerebral MRI at term equivalent age has been suggested to become part of a routine imaging protocol in PT [11]. Furthermore, advanced MRI techniques have brought great insights in PT brain development. They can explore the brain’s microstructure and connectivity [12,13]. At a microstructure scale, diffusion tensor imaging (DTI) provides maturational informations, especially the axonal myelination. At a macroscopic scale, tractography maps physical WM fibers connections, whereas functional MRI assesses neuronal networks beyond anatomical structures. The former is defined as structural connectivity and the latter as functional connectivity. Micro and macro structures are closely related to determine cerebral organization. 

Prematurity leads to increase susceptibility of the brain at a microstructural and connectomic scales. Furthermore, such disruption is exacerbated by multifactorial aggressions (inflammation, pain, etc.) that stray the brain from its initial developmental trajectory [14,15]. However, it can be counter-balanced and even reversed by neonatal interventions based on developmental care [16]. The cerebral remodeling due to external stimuli defines the concept of plasticity.

This narrative review seeks to describe normal fetal brain development, the principles of MRI techniques to assess microstructure and connectivity, and how these techniques can help to better understand the impaired brain development in PT. 

## 2. Materials and Methods

Keywords or MeSH terms were combined in a PubMed search for this review. A first request (“Magnetic Resonance Imaging” [Title/Abstract] and “brain injur *” [Title/Abstract] and “premature *” [Title/Abstract]) or ((“Magnetic Resonance Imaging” [MeSH Terms] and “Brain Injuries” [MeSH Terms] and (“Premature Birth” [MeSH Terms] or “infant, premature” [MeSH Terms] or “infant, extremely premature” [MeSH Terms])) identified 214 relevant articles. An additional 17 articles were identified by this second query: (“Magnetic Resonance Imaging” [Title/Abstract] and “brain injur *” [Title/Abstract] and “premature *” [Title/Abstract]) or ((“Magnetic Resonance Imaging” [MeSH Terms] and “Brain Injuries” [MeSH Terms] and (“Premature Birth” [MeSH Terms] or “infant, premature” [MeSH Terms] or “infant, extremely premature” [MeSH Terms]) and (“plasti *” [All Fields] or “connective *” [All Fields])). Most relevant or most cited articles were selected and so some studies were carefully excluded.

As a narrative review, authors did not deem it necessary to follow PRISMA guidance. 

## 3. Normal Cerebral Development and Neuropathology of Preterm Brain Injuries

### 3.1. Brain Development during the Third Trimester of Gestation

Many cellular processes take place during the third trimester of gestation—late neurogenesis up to the 25th gestational week, late neuronal migration, gliogenesis, synaptogenesis, myelination, and selective apoptosis [17]. Furthermore, a transient cortex called subplate is involved in the neurons’ guidance during their migration [18]. From the ninth gestational week to the first year of age, subplate and cortex successively exercise the same functional role, connecting one over another cortical areas, deep grey matter (GM) and the spine. The functional transition from the subplate to the cortex is a crucial process, during which any upset could trigger many neurodevelopmental disorders [19]. Subsequently, thalamo–cortical circuits arise, involving the motor and sensory cortex, followed by the associative cortex. As early as mid-gestation, cortico–cortical and thalamocortical fibers undergo a synaptogenetic process in the cortical subplate. Eventually, the subplate vanishes by apoptosis after transferring the first loops to the cortex.

These processes are partly genetically determined and partly dependent on sensorial input and experience [20].

### 3.2. Prematurity: High Risk of Brain Injuries

Several types of brain damages have been identified in PT [8]. 

Large parenchymal defects, caused by infarction and hemorrhage, are usually associated with intra-ventricular hemorrhages. Those damages are mostly encountered in most immature and unstable infants. Approximately half of those children are exposed to serious neurologic adverse outcomes (cerebral palsy, lower IQ) or death [21,22]. On the contrary, parenchymal infarctions of Volpe, triggered by a venous thrombosis, display more variable outcomes depending on their size and location [23].

The periventricular WM constitutes a highly vulnerable area in PT. Oligodendrocyte precursors and mitochondria are particularly sensitive to insults—inflammation, anoxo-ischaemia, deficiency in neurotrophic and growth factors— leading to a myelination disruption [24,25]. Periventricular leukomalacia used to be normally described as large, cystic necrosis associated with surrounding myelination disorder. However, cystic forms account nowadays for less than 1% of WM injuries, usually resulting in cerebral palsy [26]. Since the 90s, along with changes in neonatal practices and generalization of conventional MRI, cystic lesions have been eclipsed by multifocal WM lesions involving nodular leukomalacia in favor of a gliosis process [1,9,10]. 

Moreover, besides WM lesions considered as “qualitative” injuries, “quantitative” impairments have been described, suggesting that in utero sequence, leading to premature birth, impacts the whole process of cerebral maturation. Without overt lesions, brain metrics, measured in two dimensions or in volume after segmentation, are significantly decreased in infants born preterm, compared to control children, indicative of an impaired growth [22].

The cellular loss and WM injuries are entangled with diffuse and distant alterations in neuronal organization involving the cortex, the subplate, and thalami, defining thus the global concept of encephalopathy of prematurity [8]. It is a complex amalgam of primary destructive disease and secondary with maturational, and trophic neuronal or axonal, disturbances. Indeed, the shift towards an ex utero environment influences many mechanisms of the developing brain involving the synaptogenesis, synapse pruning, cortical organization, rise of thalamo–cortical and cortico–cortical circuitry, and programmed cell death [27,28,29]. In this regard, plasticity defines a process of cerebral remodeling to adapt to its environment or interventions. DeMater and al. introduced the concept of “double edged sword” [30]. On the one hand, neonatal interventions such as developmental care constitute positive stimuli and potential neuroprotective factors that enhance resilience of the brain [16,31,32]. On the other hand, noxious events (inflammation, lack of natural sensorial input, nociception, pro-apoptotic drugs …) cause a maladaptive plasticity of the brain, which is hence discarded from its physiological route [14,15,33,34,35].

## 4. MRI Methods: A Multiparametric Approach of Brain Maturation, Structure, and Function

### 4.1. Conventional MRI

The signal intensity of the WM on conventional sequences follows the maturation of the brain, as the contrast lies on water and fat content in the brain’s structures. Compared to adults, infants’ conventional MRI sequences show reversed signal intensities in WM and GM (from 0 to 6 months old). In T1-weighted images WM intensity appears lower than the GM, and vice versa in T2-weighted images [36]. As the cerebral maturation evolves with a decrease of water and enhances the myelination process, signal intensities change over time. 

However, at term, conventional MRI already features key markers of myelination. The myelination of the cortico–spinal fascicles may be followed as a high signal in T1-weighted sequences from the central sulcus to their projections towards the midbrain. The presence of a physiological signal in the posterior limb of the internal capsule (PLIC) is critical to confirm a myelination process [37]. Apart from an impaired myelination in PT, MRI can also identify WM injuries. Cystic periventricular leukomalacia is well diagnosed by both ultrasound and MRI. On the contrary, punctuate WM lesions representative of gliosis are only delineated by MRI scans as ischemic signals and less frequently hemorrhagic [11]. Finally, diffusive and excessive signal intensity of the WM may also be described in T2-weighted images. No histological substrate has yet been found.

In routine, the assessment of brain maturation on T1 and T2 sequences remains qualitative, and hence dependent on a radiological expertise. Quantitative T1 and T2 signals have been studied to measure brain’s maturation [38]. However, they require additional sequences and regional reference values [36].

### 4.2. Diffusion Weighted MRI

#### 4.2.1. Diffusion Weighted MRI as a Probe of Microstructural Assessment of WM and GM

Diffusion weighted MRI technique measures Brownian motion of water molecules. In brain tissue, the motion of water is not random, as it is constrained by structural obstacles, such as axonal membranes. Therefore, water molecules diffuse more freely along the direction of axonal fascicle. This directional dependance of water molecules movement is termed anisotropy. For DTI acquisition, diffusion gradients are applied in more than six directions to use a tensor diffusion model. For each 3D pixel, called a voxel, several quantitative parameters can be calculated from this tensor model: fractional anisotropy (FA) (overall directionality), mean diffusivity (MD), axial diffusivity (diffusivity along the main tensor axis), and radial diffusivity (diffusivity perpendicular to the main axis) can be derived from the tensor model (Figure 1) [39]. These parameters quantitatively assess the water diffusion properties in a voxel reflecting brain microstructure including WM and GM maturation. Along with maturation, they follow cellular milestones. For instance, FA increases in the WM, together with oligodendrocyte extension, glial cell proliferation, and pre-oligodendrocyte ensheathment [36,39]. On the contrary, FA decreases in the cortex. Immature cortical plate is initially anisotropic due to radial orientation of glia fibers and apical dendrites, and becomes isotropic at term equivalent age due to dendritic arborization [36].

#### 4.2.2. DWI Tractography and Structural Connectivity

Using DTI, directionality maps showing the main direction of the diffusion tensor for each voxel can be modelized, demonstrating the organization of main WM bundles. 

Different 3D reconstruction tools can be used to assess the main direction of WM bundles in 3D. However, these tools, which are based on simple DTI considering a single fiber population per voxel, must be subject to caution due to the occurrence of false negatives (premature termination of a tract) or false positives (switch of another bundle crossing the same voxel) [40,41]. More complex tools can be used to deal with crossing fibers, but they require the acquisition of longer DTI sequences.

Structural connectivity refers to networks that anatomically interconnect brain regions. It is typically measured in vivo in humans using diffusion weighted imaging tractography by combining 3D reconstruction of fiber bundles and quantification of microstructural characteristics of those bundles using diffusion metrics (FA, MD, radial, and longitudinal diffusivities) (Figure 2).

### 4.3. Functional MRI, Functional Connectivity

Functional MRI aims to describe brain activity based on the hemodynamic response following neuronal activity using blood oxygenation level-dependent (BOLD) sequences. The local imbalance between oxygenated and deoxygenated hemoglobin during neuronal firing generates a local magnetic field change that creates a contrast in MRI [42]. Functional MRI is therefore used to probe brain activity during specific cognitive or sensorimotor tasks. In addition, it has been shown that synchronized low-frequency fluctuations of BOLD signal can be identified at rest in remote brain regions belonging to the same network (resting state functional MRI). This is the basis of the study of functional connectivity that reflects the functional architecture, i.e., the association of brain regions within long distance networks, as well as the relationships between different networks. Hence, networks identified in resting state functional MRI (rs fMRI) are named resting state networks (RSN). Resting state MRI is suitable for newborns imaging studies as it does not require patient cooperation, and can be performed during sleep. 

Finally, older children could benefit from task-based MRI. It explores the effective connectivity, and may bring great insights to further understand the mechanisms of plasticity. Effective connectivity assesses the causal influences that neural units exert one over another [43]. It reflects the functional connectivity during a dynamic task.

### 4.4. Brain Connectome

Interestingly, while there is a large overlap between structural and functional connectivity, there is no strict correspondence between them. 

Historically neuroscience stated the association between a brain region and a specific function. In other words, brain organization is based on the segregation of brain regions with a task specifically linked to an anatomical region. 

In addition, more recent approaches highlighted that brain function relies on large scale functional networks composed of interactive regions, sometimes remote, revealing an integrative organization.

Brain network can be modeled accordingly following the graph theory, including an ensemble of neuronal elements termed as “nodes”, connected one to another by “edges or links”. Ideally, nodes should represent brain regions [12,44]. Links may correlate to WM tracts (structural connectivity) or to a functional circuitry (functional connectivity). Different metrics can be used to define the network organization and its general geometry or topology. 

A “segregation organization” is opposed to a “network integration organization” (Figure 3). In a “network segregation”, one module contains several nodes densely linked one to another, but sparsely connected to other nodes belonging to a different community. It reflects a potential specialized function of the network. On the contrary, a network integration demonstrates a competence to travel information through different brain regions (nodes) [45].

## 5. MRI Studies of Early Typical Brain Maturation, in the Third Trimester of Gestation

### 5.1. Maturation of WM

Anisotropy does not exclusively reflect a myelination process. It also relates to the development of oligodendrocytes and axonal density. A total of three major steps occur during the maturation of the WM [39]. First, fibers inside a bundle undergo a fasciculation phase, where oligodendrocytes line up. Second, these oligodendrocytes proliferate during a pre-myelination stage. Meanwhile, water concentration decreases as membranes densify. Finally, the myelination ensheathment of the axon takes place with decreased membrane permeability and extracellular distance between membranes. Together with the maturation, anisotropy (FA) increases, whereas mean and transverse diffusivities decrease. Longitudinal diffusivity first increases and then decreases. 

Based on histological explorations, myelination follows a central to peripheral and posterior to anterior gradient up to three years old [46]. Diffusion tensor imaging studies of in utero fetuses and PT have confirmed such time course. Between 28 to 43 weeks of gestational age, WM in cerebral peduncles, internal capsule and commissural tracts of corpus callosum (CC) were the first to emerge [45]. After birth, PT and infants’ models, designed by Dubois and al., described how myelination successively progressed in the first months— (1) motor and somatosensory pathways of the cortico–spinal tract (CST), (2) spinothalamic tract and fornix, (3) visual tract of the optic radiations, arcuate and inferior longitudinal fascicles, and (4) anterior limb of the internal capsule and the cingulum [47]. 

Maturation is asynchronic, but also asymmetric. For instance, speech processing is related to the left perisylvian region. Moreover, human species are distinguished by a lateralized hand preference. Such asymmetry is already set early in life. As early as the first three months of life, the auditory areas of the left hemisphere have already showed a predisposition for the language processing [48]. Furthermore, at a microstructural scale, a leftward asymmetry in the arcuate fasciculus and CST was demonstrated with a higher FA compared to the contralateral side, as soon as one month of age [49]. Those two bundles are involved in the language and sensory-motor networks. However the correlation between the asymmetry of motor tracts and handedness remains debated. Moreover, correlation between structure and function is not systematic and remains one major challenge in neuroimaging of cerebral connectome studies.

### 5.2. Maturation of the Cortex

The cortex matures simultaneously with the development of its corresponding WM as shown by Smyser et al. [50]. On conventional MRI sequences, rolandic sulcus and CST as well as the thalamostriate pathways are easily visible at term. However cortical microstructure is more complex to analyze. It is constrained by the development of dendritic arborization, synaptogenesis, and myelination of intracortical WM. The cortical thickness and the complex cortical folding pattern also render the evaluation difficult. In DTI, GM maturation was characterized by a decrease of MD and FA as opposed to WM [51]. The diminution of MD was due to the increasing cellular density and complexity, whereas the maturation of dendritic architecture and neurite outgrowth led to a decreased FA.

Furthermore, other quantitative MRI modalities have confirmed DTI and postmortem studies [52]. Primary sensorimotor regions myelinated earlier followed by the primary visual cortex around the calcarine fissure, primary auditory cortex, and somatosensory cortex [51,52]. Such hierarchy remains over the infancy period.

### 5.3. Maturation of Structural and Functional Connectivity

Structural connectivity has been studied in vivo as early as 27 weeks of post menstrual age [53]. The vast majority of the large pathways present in an adult connectome was also identified in neonates [54]. In term of topology, they were already exhibiting, before term, a small-world architecture, characteristic of the adult organization. In such architecture, networks are clustered as opposed to random, which enables to process the information more efficiently. Besides, hubs, i.e., highly connected nodes, were described in the cingulate cortex, superior frontal and parietal regions, and left precentral and post gyrus. Furthermore, brain regions encountered a gradient in maturation [55]. Between 27 and 45 weeks of postmenstrual age, connections between primary sensorimotor, occipital, and frontal cortex featured the highest rate growth, especially short connections within the same hemisphere [51,53,56]. Eventually, during the prenatal stage, the structural network would evolve from a segregation of locally efficient groups of regions to an integrative organization with the development of long-range connections [56,57].

It is assumed that functional connectivity leans on an effective structural connectivity, even if it is not exactly correlated to it. Resting state networks were detectable by the third trimester, but rather at an immature state compared to structural connectivity [56]. Primary networks, such as the sensorimotor, visual, and auditory networks, and thalamocortical connectivity could be detected in PT as early as 26 weeks of gestation [50,58]. Maps revealed a bilateral symmetry between hemispheres in favor of a qualitative homotopic functional connectivity. In other words, in early gestational age, strong connections remained inter-hemispheric, and became intra-hemispheric over time. Certain higher-order networks, such as the default mode network and salience network, were also delineated very early on. The salience network includes nodes in the amygdala, hypothalamus, ventral striatum, thalamus, and specific brainstem nuclei [59]. It is implicated in the integration of sensory, emotional, and cognitive information. Default mode network is built on the posterior cingulate cortex and ventral anterior cingulate cortex, and is activated during resting states [60]. 

As neonatal functional architecture greatly overlaps with the adult’s connectome, it should constitute the proto-architecture that would refine along the post-natal life [54]. Graph theoretical model enabled to describe similar hubs and rich club architecture as early as 31 weeks of post menstrual age in primary motor, somatosensory, visual, and auditory regions [61]. A rich club organization described well-interconnected hubs, which has been proven to be efficient for communication within the brain. Along with gestational age, the organization evolved with more efficient local connectivity, as well as the development of long-range connections. Finally, similarly to structural connectivity the balance between a segregation and integration organization favored the former [56].

## 6. Brain Alterations in PT

### 6.1. WM Microstructural Alterations

#### 6.1.1. Cerebral Injuries and Risk Factors of WM Alterations Disruption

Brain lesions may disrupt the underlying periventricular WM bundles [21,62,63,64,65,66,67]. A recent prospective study included PT sustaining brain injuries compared to uninjured PT and full-term infants [67]. Altered WM with reduced FA was found in the PLIC, the right cingulum, the lentiform nuclei, and the CC. Higher MD was also measured in both optic radiations and cingulum due to increased edema and lower cellularity. Interestingly, one the one hand, disruption of the WM was predominantly located in the caudate nucleus and CC in intra-ventricular hemorrhages. On the other hand, post hemorrhagic ventricular dilatation was related to alteration of the internal capsule and optic radiations [63,67]. Nodular periventricular leukomalacia was also associated with lower FA values in the PLIC, cerebral peduncles, corona radiata, and arcuate fasciculi, along the CST [62,64,65]. The load of punctuate lesions was negatively proportional with FA.

Finally, all factors causing a systemic inflammatory state such as broncho-pulmonary dysplasia, infections, necrotizing enterocolitis, and intra-uterine growth restriction, constituted risk factors to a disruption of the WM microstructure [41,66,68,69]. Number of days of parenteral nutrition and mechanical ventilation, higher social risks, and male sex also predisposed WM alteration [66,69].

#### 6.1.2. Alteration of the WM Microstructure in PT without Apparent Brain Injuries

Even in the absence of obvious brain lesions, the preterm period, as an early ex utero life extraction, would independently cause a disruption of the cerebral microstructure. Literature is relatively vast, yielding various results. In PT, an overall alteration occurred in a wide range of WM bundles known to play a role in motor, language, reading, or mathematic skills [41,70,71,72,73]. Among all the anatomical sites, the CC was widely studied as a major inter-hemispheric track vulnerable to ischemia and inflammation, showing a reduced FA which interestingly raised the hypothesis of a less efficient inter-hemispheric transmission [41,70,71,72,73]. Moreover, Li and al. gathered studies in a meta-analysis comparing PT versus term infants through different periods of life: term equivalent age, infancy, childhood, and adolescence [70]. The decreased FA of numerous WM fascicles persisted over every period of the timeline [40,70,71,72]. One should bear in mind that FA values need to be interpreted according to the age of PT. In the early infancy myelination has not occurred neither in the splenium nor in the genu of the CC. A fall of FA might then be related to a delay of the pre-myelination stage with lesser pre-oligodendroglial cells, and disorientation along the axonal axis [39]. 

Finally, at the cortical level, FA also declined in PT, along with the maturation and hence gestational age [51]. However, the speed of drop was reduced in PT with low gestational age. At term-corrected age, PT showed higher FA than term infants.

#### 6.1.3. Correlation of WM Abnormalities with Neurodevelopment

The widespread disruption of WM involves numerous bundles which share multiple roles in both motor and cognitive development. Consequently, their disturbance at a microstructure scale have been related to disabilities [51,74]. Prematurity and WM lesions both acted as independent factors.

A slower maturation, i.e., a slower increase of FA during the preterm period, and a lower FA at term equivalent age in the basal ganglia, CC, and PLIC, were associated with a worse motor and cognitive outcome at 18–24 months [63,74,75]. Griffiths’ developmental quotient also positively correlated with the maturation rate of FA in regions of interest placed in frontal, parietal, and occipital cortex [51].

Such link was confirmed by long-term follow-up. At seven years old higher MD, axial and radial diffusivity, in the cerebellum, and the inferior occipital region, were related with both motor and cognitive impairments [64]. Intellectual quotient and reading skills were positively associated with FA in the occipital CC at age seven [41]. Reduced FA in the CC was also linked to impaired attention performance, and specifically in orientation scores. Finally, a strong correlation was shown between a reduced FA in the left cingulum, the superior longitudinal fasciculi and attention, orientation, or alerting, performances [71].

At adolescence, the same pattern between WM disruption of those identical bundles remained in relation with poor executive function [73]. PT teenagers with mathematic difficulties had lower neurite density and lower FA in the corona radiata, internal and external capsule, and the left fronto–occipital fasciculus [76].

### 6.2. Abnormal Structural Connectivity in PT

#### 6.2.1. Cerebral Injuries and Structural Connectivity

Tractography finely illustrates in 3D how WM tracts are damaged by brain lesions [12,77]. In PT with unilateral periventricular hemorrhagic parenchymal infarctions, the subsequent WM loss reduced the efferent CST volume. Moreover, those patients showed a contralateral hemiparesis as early as the first year of age. However, the afferent thalamo–cortical tract was relatively spared as it could circumvent the area of focal WM injury [77]. In some patients, the thalamo–cortical tract delineated two routes to link the thalamus to the perirolandic region, at term equivalent age. The first one connected the thalamus to the insula area, and the second one the insula to the perirolandic region. Interestingly, this double pathway was no longer found at one year, but rather consolidated into one single tract. As the CST matures rapidly especially in the early third trimester of gestation, a compensatory mechanism would be limited. By contrast, thalamo–cortical afferences still undergo a maturation process through the subplate to reach the cortical layers. It allows to take advantage of a potential plasticity and establish a compensatory route. 

Moreover, Smyser and al. have represented the benefit of ventriculoperitoneal shunt in post-hemorrhagic ventricular dilatation [12]. Tractography illustrated how, in one week after the ventriculoperitoneal shunt, CST of the less injured hemisphere had grown significantly in a compensatory manner, whereas the contralateral CST which was more severely damaged remained small.

Interestingly, in parieto–occipital leukomalacia, the visual WM pathways, i.e., the optics radiations, the ventral visual stream (inferior longitudinal fascicle, and inferior fronto–occipital fasciculus) and the posterior thalamic radiations displayed an abnormal connectivity [78]. However, similarly a reorganization occurred in the fronto–striatal and fronto–limbic pathways with multiple nodes showing higher efficiency, acting like a compensation.

#### 6.2.2. Structural Connectivity Alteration in PT without Apparent Brain Lesions

The disruption of the structural connectivity by the prematurity has coincided with a decrease of the cerebral volume and an altered microstructure [79,80,81]. It prevailed in the thalamo–cortical structural network associated with a reduction of the thalamic volume [82]. The structural connectivity remained impaired at term equivalent age. 

In term of organization and topology, connections with a fast growth passing through the deep GM were preserved by the prematurity [51,55]. Furthermore, between 30 and 40 weeks of gestation, connections between core hubs kept proliferating in order to establish strong networks at term. Overall, PT favored a segregation organization and a rich club architecture, a core of local highly connected organization. However, its existence seems to be detrimental to the brain regions, with a slower physiological growth [51]. Indeed, local connectivity, involving the cerebellum and thalamus, was the most impacted by the degree of prematurity as they showed less efficient connections [55].

This disrupted pattern seems to persist over time [44,83]. From 1 to 3 years of age, the fronto–temporal region including the anterior centrum semi-ovale, corona radiata, and the genu of CC, would normally undergo a critical development in terms infants. Yet, it was not the case in PT who still encountered a widespread reduced connectivity [83]. Differences also remained at six years in the frontal region and the limbic system. Moreover, a rich club architecture and a segregation organization prevailed even at that age over a normally integrative pattern [44,84]. As a consequence, PT might lack the ability to exchange information between different brain regions. Interestingly, in the same cohort of six years old children, intra-uterine growth retardation was a factor of connectivity disorder.

Finally, such disruption was confirmed in young adults [81]. Networks implicated in somato-sensory functions, i.e., precentral gyrus, mid temporal area, deep GM, and cerebellum pathways, remained more organized locally than in term children keeping the same pattern as described above.

#### 6.2.3. Structural Connectivity and Outcome Prediction

Tractography 3D modeling of damaged CST is an intuitive and educative way of predicting motor outcome. However, conventional imaging already demonstrates good specificity and sensitivity for motor prognosis [85]. One purpose of connectivity research is to correlate structural and functional networks to cognitive disorders. In this regard, studies are relatively scarce. Bayley score at two years was related to thalamocortical pathways integrity [86]. Moreover, performance IQ of WPPSI scale was significantly associated with markers of integration and segregation networks [87].

### 6.3. Functional Connectivity Alteration

#### 6.3.1. Functional Connectivity Disruption by Brain Injuries, and Plasticity

As opposed to structural connectivity, the functional connectivity’s disruption due to a brain lesion does not systematically reflect the damage of the WM tract itself. It may also reveal indirect consequences of WM injuries. Diffuse excessive hypersignal intensity described on conventional MRI was associated with aberrant connectivity in executive and fronto–parietal RSN, which were linked to executive functions and language processing [88]. Moreover, severe intra-ventricular hemorrhages or moderate–severe periventricular leukomalacia impacted RSN located in the thalamus, motor, auditory, visual, cortices, and the cerebellum [89]. The thalamo–cortical connectivity was also disrupted proportionally to the severity of the injury. In this regard, inter-hemispheric (long range connectivity between homotopic counterpart) and intra-hemispheric (short range connectivity within the regions of interest) connections were decreased compared to control. 

Along with the altered thalamo–cortical connectivity in WM injuries, connectivity was increased in the salience pathway [61]. Interestingly, this pathway was implicated in the “global neuronal workspace” circuit, which was responsible of neonatal consciousness and sensory process [90]. Altered thalamo–cortical connectivity might also be compensated by a higher inter-hemispheric connectivity, especially in peri-rolandic and insular regions [77]. It was demonstrated in PT who sustained less severe, peri-ventricular hemorrhagic infarctions, and leukomalacia [61,77,90]. The preservation of the inter-hemispheric connectivity has been playing a key role in plasticity to guarantee a better neurodevelopmental outcome. Interestingly, loss of inter-hemispheric functional connectivity in post hemorrhagic ventricular dilatation has been rapidly reversible after ventriculo–peritoneal shunt [12]. Finally, in some cases, functional connectivity in GM within the injured area was spared or otherwise increased [61,89]. This provided evidence of another compensatory mechanism. Functional WM disruption is a complex ensemble of direct and collateral effects, involving respectively mono-synaptic and polysynaptic connections.

#### 6.3.2. Functional Connectivity Disruption in PT without Apparent Brain Lesions

The disruption of the functional connectivity observed in PT at term equivalent age involves the thalamo–cortical and cortico–subcortical pathways [58,91,92]. A significant reduction was demonstrated between thalamus, sensorimotor cortex, brainstem, and cerebellar vermis [58]. Moreover, connections between basal ganglia nodes and frontal cortex nodes were found to be less prominent in PT compared to term infants [91]. 

Furthermore, PT displayed less long-range connections [58]. Interhemispheric connections were also less robust.

In addition, PT infants showed a dense locally connected organization [93]. A core set of nodes may be identified in posterior cingulate cortex, inferior parietal, temporal lobes, and visual cortex, but featured a reduced strength compared to term infants, with a decreased connectivity between hubs, and a less functional segregation.

Finally, functional disruption persists over time [94]. At 18 and 36 months old, functional connectivity was stronger within the visual areas among PT, whereas term infants displayed a stronger connectivity between motor and both visual and frontal regions. Such dysconnectivity between visual and motor areas was confirmed in adolescence, as well as in the dorsal attention network [95]. In contrast, higher connectivity was found between the sensory motor network and the central executive and salience networks. Interestingly, the hyperconnectivity suggested several hypotheses by the authors: first, a plasticity mechanism to cope with the counterpart hypoconnectivity, or conversely a WM disruption effect, especially the lack of synaptic pruning.

#### 6.3.3. Functional Connectivity and Outcome Prediction

The predictive value of functional disruption on neurodevelopment needs further studies. However, ex PT children and adults with cerebral palsy displayed a reduced connectivity of the motor cortex towards the bilateral paracentral lobule and the cingulate motor region, the thalamus connections to the caudate nuclei and the cingulate cortex and, between the sensori-motor and visuo-motor pathways [96]. Furthermore, describing how functional and structural connectivity may interact with each other proves instructive [97]. Compared to control, children and young adults with cerebral palsy exhibited in the whole brain a reduced efficiency at a structural scale, but also a rather intact functional network. Such results rose the hypothesis of a functional reorganization. The opposite pattern was demonstrated in the motor network as the structural connectivity was preserved whereas the functional one showed reduced global efficiency. Hence, authors have raised the hypothesis of a structural reorganization. It is difficult to conclude which mechanism is more critical for the neurodevelopment.

## 7. Negative and Positive Plasticity of the Brain during the Neonatal Period

As the brain undergoes a critical developmental process during the neonatal period, it is further influenced by exterior stimuli. Nowadays, neonatal therapeutic strategies are based on developmental care. They aim to minimize stressful stimuli from the environment and promote sensory stimulating care. In this regard, negative and positive experiences impact cerebral microstructure and connectivity.

### 7.1. Stress and Nociceptive Stimuli

Any event that engenders an over-stimulation for the newborn would be considered stressful. Therefore, stress encompasses a large variety of stimuli. It ranges from changing the diaper to painful procedures. 

Pain overstimulates vulnerable and immature neurons, through excitotoxic cellular damage with excess of calcium and glutamate release [98]. Axonal development alters subsequently. Physiologically, thalamo–cortical and limbic pathways play an important role in nociceptive encoding [99]. Indeed, thalamus conveys nociceptive signal to cortical areas, including the somatosensory and limbic cortices. As for the brainstem, the limbic system and the hypothalamus are responsible for a feedback loop that modulates the spinal dorsal horn response to an external stimulus. It allows to refine sensory inputs as tactile, thermal, or painful sensation. Such development process has been lacking during the neonatal period [14,15]. Therefore, PT have been subjected to a central sensitization to repeated painful stimuli without inhibitory feedback, which subsequently disrupted the microstructure and the connectivity. Furthermore, in early life the descending pathway regulation to the spinal activity is more likely conducive to “lower threshold” tactile stimuli. By contrast, inhibitory retro-control to “high threshold” painful input occurs later in life, in particular after the preterm period. Authors have also postulated that such a process was compromised by prematurity [14,15].

At a microstructure scale, nociceptive stimuli (skin breaks) might affect the anisotropy diffusively, including basal ganglia, thalami, optic radiations corpus callosum, and temporal lobes [33,34,35]. At a structural scale, pain was associated with a thalamic volume loss in concordance with an altered connectivity in thalamic and sensory motor RSN [14,15]. A decreased functional connectivity of the insula and the limbic system was also demonstrated, including connections between the right amygdala and the ipsilateral hippocampal and parahippocampal, regions [15]. Moreover, a weaker functional interhemispheric connectivity from the right temporal cortex was found.

It is worth noting that glucose suction did not have any effect positive effect on the WM disruption, and that the impaired connectivity concerned PT at an early age rather than at term corrected age [15,35]. 

Finally, as early as in the prenatal period, anxiety and depression of pregnant women perturbed the amygdala’s functional connectivity to the brainstem, the thalamus, and the hypothalamus with a reduced network’s strength [100]. As the amygdala connectivity is regulated through GABAergic transmission, authors have suggested that stress disrupted the migration of GABAergic progenitors and impaired synaptogenesis.

### 7.2. Positive Stimulation and Musicotherapy

Musicotherapy is used to relieve stress in neonatal care, and may improve a newborn’s well-being [31]. Musicotherapy promotes maturation of tracts involved in auditory and socio-emotional processing [16]. Indeed, compared to PT without any interventions, PT with musicotherapy showed higher FA in acoustic radiations (which transmit auditory information from the thalamus to the primary auditory cortex), uncinate fasciculi (which participate at emotion’s encoding), and in the external capsules. 

The beneficial plasticity of microstructure matches with functional connectivity results [101]. Salience RSN connects modules implicated in sensory perception and cognition. Musicotherapy enhances connectivity in those modules that involve the auditory, sensorimotor, superior frontal, thalamus, and precuneus networks. 

Therefore, musicotherapy is a promising tool to prevent the detrimental effect of prematurity on the cerebral maturation and organization.

## 8. Conclusions

Premature birth deprives the brain from its physiological in utero environment, during a critical period of intense development. Prematurity is disruptive at a microstructural and connectomic scales, which constrain the brain maturation and organization. Subsequently, aberrant circuitries are reshaped. Furthermore, a locally connected organization persists over the time precluding to evolve towards a long-distance integrative architecture. 

Nevertheless, the PT brain shows resilience and plasticity, driven through an interhemispheric connectivity and alternative WM routes. Both occur in structural and functional networks. In this regard, plasticity is assumed to be positive and reflects a compensatory mechanism. However, the restructuration to an atypic circuitry could also be interpreted as a “maladaptive plasticity”. Indeed, plasticity is a process based on experience that nurtures and shapes the brain. This occurs as early as the neonatal period, and can be beneficial or detrimental. More interestingly, part of the experience is directly derived from neonatal interventions of care providers, which bring new perspectives in the management of PT.

Cerebral MRI constitutes a versatile and efficient means of investigation to exhaustively explore the brain. Effective connectivity studies go beyond the scope of this review as well as gyrification and volumetry. However, those are also influenced by prematurity and plasticity factors [102,103,104]. Non-invasive neuroimaging techniques, such as magnetoencephalography which measures neural activity may also be coupled with MRI [105]. Finally, deep learning has been introduced in various radiological applications and brings new promises [106]. In other words, the field of research and application of brain MRI is wide. Its evolution follows the strides in computing and data processing. Thus, the perspectives seem endless, and give MRI a key role in neonatal neuroscience. The challenge remains to correlate those invisible alterations to subsequent neurodevelopment, especially the actual rise of cognitive and executive function disorders. As a consequence, methodology in MRI assessment is critical, and further longitudinal studies are needed before drawing any recommendations for the neonatal plan of care.

## Figures and Tables

**Figure 1 children-09-00356-f001:**
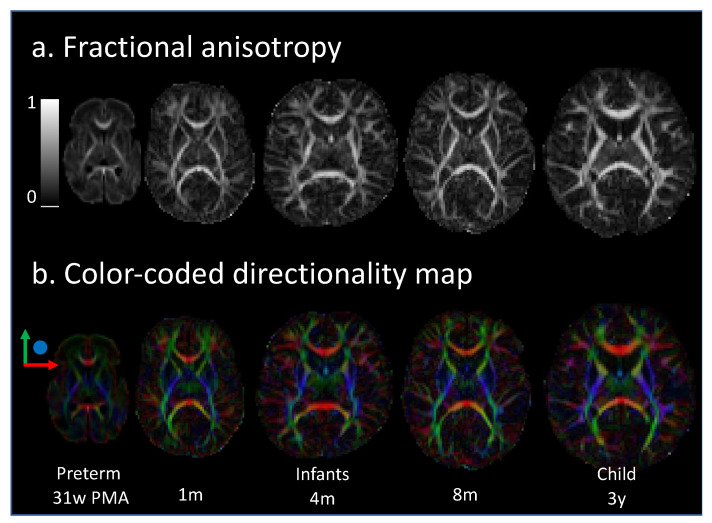
Evolution of DTI maps during development. Fractional anisotropy (**a**), and color-coded directionality (**b**) maps are illustrated according to the age—31 weeks of postmenstrual age, at 1, 4, and 8 months of age, and 3 years old. Adapted from ref. [36].

**Figure 2 children-09-00356-f002:**
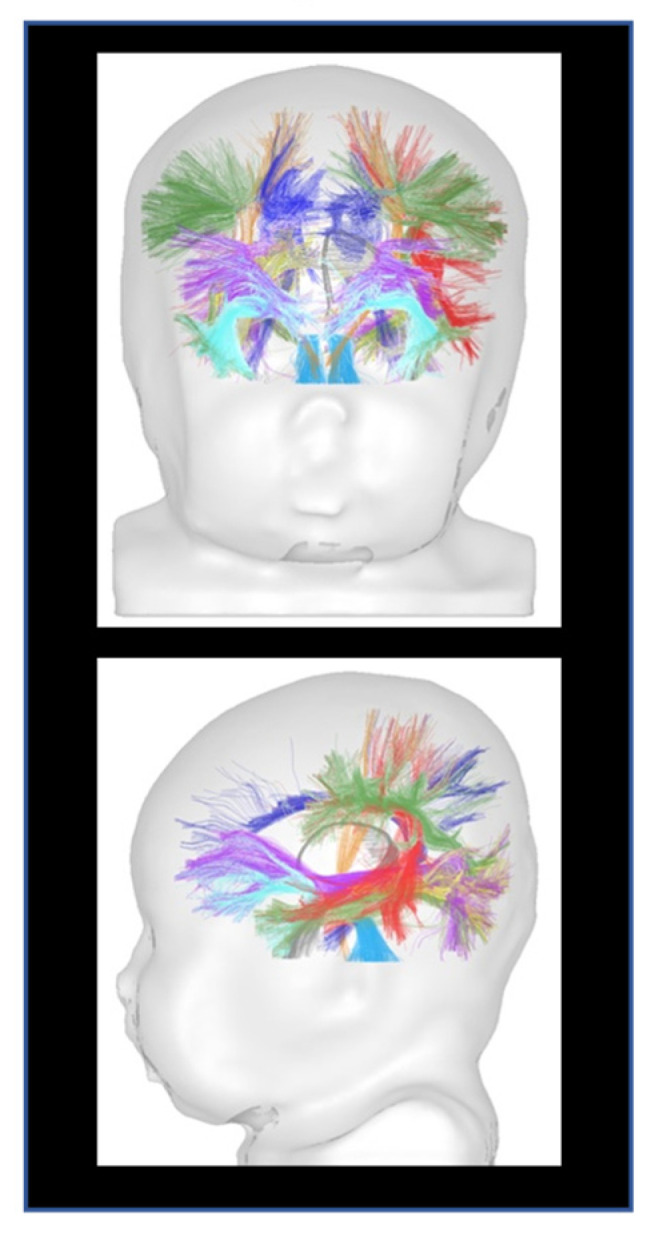
Tractography of WM bundles in a 1-month-old infant. The trajectory of the main WM bundles (projections, callosal tracts, limbic, and associative bundles) can be generated. Adapted from ref. [36].

**Figure 3 children-09-00356-f003:**
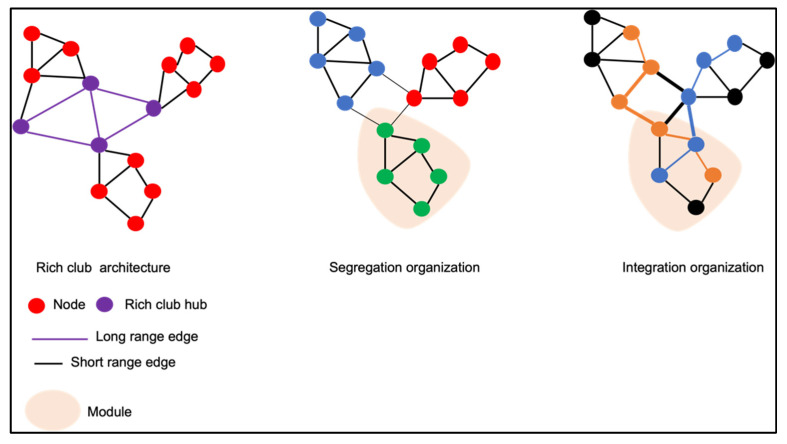
Illustration of a cerebral network according to the graph theory: nodes are connected one over another by edges. Nodes which play a significant role in exchanging information are hubs. Well-interconnected hubs belong to a rich club. In a segregation organization, modules represent networks with dense connections between nodes within the same module but sparse connections between nodes of other modules. In an integration organization, disparate brain modules are well-interconnected.

## Data Availability

Not applicable.

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
