# Peer review of "Advanced Brain Imaging in Preterm Infants: A Narrative Review of Microstructural and Connectomic Disruption"

_children, 2022, doi:10.3390/children9030356_

Round 1

Reviewer 1 Report

A wonderful comprehensive review.

Author Response

Dear reviewer,

We sincerely thank you.

Best regards,

P Vo Van 

Reviewer 2 Report

The reviewed manuscript has an extensive literature review of advanced brain imaging in preterm infants. This is a fascinating review. The review is well written, and included references are clear and well-reasoned.  I have checked most of the references but not all due to time constraints. However, it is not clear whether entire searched literature is included or some studies are carefully excluded.This is a narrative review so I am not expecting any underlying rationale for the review.  However, author didn't clearly concluded the findings from literature review. Research perspective of this review is not clear.  Any schematic reperesentation would be more interesting and helpful to represent  how this huge body of knowledge would help future research studies. The review is bit lengthy it would be helpful for readers if author can reduce some words.

Author Response

Dear reviewer,

We thank you for your valuable remarks.

In the methodology’s paragraph, it was unclear whether the entire searched literature was included, or some studies were carefully excluded. In this regard we have modified this paragraph to mention our literature requests. As a narrative review, we excluded some studies that we deemed to be irrelevant. We hope that these changes clarify the methods enough.

You have asked to be more conclusive regarding the findings from the literature review, and to clarify perspectives. Therefore, we have modified these parts to address the authors’ opinion and to be more persuasive.

Unfortunately, we failed to draw another scheme to summarize the topic, especially because we lack material to represent the connectomic and the microstructure. We sincerely apologize for that.

Finally, efforts have been made to reduce the length of the article and improve English syntax

Thank you for your consideration of this manuscript.

Philippe

Reviewer 3 Report

Thank you for inviting me to review “Advanced brain imaging in preterm infants: a narrative review of microstructural and connectomic disruption.” by Philippe Vo Van and colleagues. This article is well structured and reviewed very important issues in the neonatal care. I was very pleased to read this review article.

Despite this article reviewed almost important parts of brain imaging, I would like to suggest to add more about cerebellar hemorrhage related connectomic disruption. Because one of reasons that neonatologists check brain MRI in term equivalent age is detect cerebellar hemorrhage which is rarely found in brain ultrasound. 

Minor 

The title of third paragraph was 'Normal cerebral development' but it had '3.2. Prematurity: high risk of brain injuries' and it explained several types of brain damages. Please change the title 'Normal cerebral development'.

Author Response

Dear reviewer,

Thank you for your remarks about the interesting pathophysiology of cerebellar hemorrhages.

To our knowledge, studies related to cerebellar hemorrhages and connectomic’s disuption are rare. Some very interesting studies have mentioned cerebral growth failure related to cerebellar injury, namely cerebellar cross diaschisis. However, to our knowledge, little is kown on microstructural impact of those injuries.  Only one study mentioned in the article included some patients sustaining cerebellar hemorrhages. But their number was too little to draw any conclusions. In this regard, we did not deem necessary to mention cerebellar lesions. However, cerebellum implications in the present topic are described in paragraphs 6.1.3; 6.2.2; 6.3.1.

We have also changed the title “Normal cerebral development” to “Normal cerebral development and neuropathology of preterm brain injuries” to you request.

Thank you for your consideration of this manuscript.

Philippe VO VAN

Reviewer 4 Report

This is a very extensive review of potential brain injuries that result from prematurity and complications of prematurity and its management. However, the level of English language is very concerning. Many words are used out of context and some made up words like 'favorized', 'fragilized' not in the English dictionary were used.

I suggest the authors seek the help of an English editor to review the work before submission.

Author Response

Dear reviewer,

We sincerely thank you for your comment. The manuscript has been reviewed by an American citizen to improve the English syntax.

Thank you for your consideration of this manuscript.

Philippe Vo Van, M.D

Round 2

Reviewer 3 Report

Thank you for all your efforts.

Reviewer 4 Report

The authors have addressed the quality of English language raised in my earlier review.